# Public welfare donation, rent sharing, and income gap within enterprises

**Jiantao Chen[1], Xiang Luo[2]\*, Xiao Wang[3]**

**1** School of Economics and Management, Southwest Petroleum University, Chengdu City, Sichuan Province, China, **2** School of Economics and Management, Southwest Jiaotong University, Chengdu City, Sichuan Province, China, **3** College of Economics & Management, Mianyang Teachers' College, Mianyang City, Sichuan Province, China

\* Flytigerluo@163.com

**Data Availability Statement:** All data files are available from the CSMARS database: https://cn.gtadata.com/.

**Funding:** The authors received funding support by Mianyang Social Science Key Research Base - Sichuan Modern Circulation Economy Research

## Abstract

This study utilizes data from A-share listed companies between 2011 and 2020 to empirically investigate the impact and mechanism of public welfare donations on the internal income gap of enterprises. The research findings indicate that public welfare donations significantly increase the per capita salary of management, while their impact on the per capita salary of ordinary employees is not significant, thus leading to an expansion of the internal income gap within enterprises. The results from mechanism testing reveal that the income tax benefits resulting from charitable donations and the rise in corporate operating income have contributed to an increase in excess rent shared by enterprises and employees. Due to a stronger bargaining power, management shares more excess rents, thereby widening the income gap within the enterprise. Heterogeneity analysis demonstrates that public welfare donations have a greater impact on the internal income gap of non-state-owned enterprises; however, limiting executive compensation and enhancing employees' bargaining power can mitigate this widening effect caused by public welfare donations on enterprise's internal income gap. The research value of this study is threefold. Firstly, there is a scarcity of studies on the impact of public welfare donations on the income gap within enterprises, and this study contributes to enriching the research in this area. Secondly, this paper examines the effect of tax incentives for public welfare donations on the internal income gap of enterprises, thereby deepening the research on the impact of tax reduction and fee reduction, as well as expanding our understanding of corporate income tax preferential policies. Thirdly, it offers insights into improving enterprise compensation systems and enhancing corporate governance. Senior executives can potentially allocate more excess rent through their strong bargaining power. If their compensation remains unrestricted, it may lead to a widening internal income gap and negatively affect company operational efficiency.

## Introduction

Corporate social responsibility encompasses various aspects, including economic, legal, ethical, and charitable responsibilities [1]. Public welfare donations fall under the category of

Center, Tianfu College, Southwest University of Finance and Economics (XDLTJJ2023YB11) for this work. Specific grant numbers: XDLTJJ2023YB11 Initials of authors who received each award:X W Full names of commercial companies that funded the study or authors: Mianyang Social Science Key Research Base - Sichuan Modern Circulation Economy Research Center, Tianfu College, Southwest University of Finance and Economics Initials of authors who received salary or other funding from commercial companies:X W URLs to sponsors' websites: https://mce.tfswufe.edu.cn/index.htm The funders had no role in study design, data collection and analysis, decision to publish, or preparation of the manuscript.

**Competing interests:** The authors have declared that no competing interests exist.

charitable responsibility and are considered a crucial civic behavior for enterprises, representing the highest form of corporate social responsibility [2]. The impact of public welfare donations on corporate social responsibility is reflected in: firstly, through public welfare donations, companies can improve their reputation, social status, third-party organization rankings, and fulfill their economic responsibilities [3]; Secondly, public welfare donations can enable enterprises to obtain legal and compliance recognition from external and internal stakeholders [4,5], fulfilling legal and ethical responsibilities; Finally, public welfare donations fulfill their charitable responsibilities, As the main means of three distributions, public welfare donations can be of great significance in regulating residents' income gap, alleviating social conflicts, and promoting fairness and justice. However, the role of public welfare donations in the internal income distribution of enterprises is rarely studied. As an important part of the income distribution of the whole society, the internal income distribution of the company will have a significant impact on the income distribution pattern of the whole society [6]. The existing research results show that the income gap within Chinese enterprises is gradually increasing, and its contribution to the overall social income gap has reached 33% [7,8]. In this context, exploring the impact of public welfare donations on the internal income gap of enterprises and Countermeasures to promote the fairness of domestic income distribution and achieve common prosperity in China is of positive significance.

This study aims to analyze the impact and mechanism of public welfare donations on enterprise income distribution from the perspective of the income gap between the company's management and ordinary employees. The management of the company in this study refers to the members of the board of directors, the members of the supervisory board and the senior management of the company; ordinary employees refer to the members of the company other than the management. Theoretically, enterprises get two types of excess rents, namely tax preference and operating performance, due to public welfare donations. Article 9 of the Enterprise Income Tax Law of the People's Republic of China: The portion of public welfare donation expenses incurred by enterprises within 12% of the total annual profit is allowed to be deducted when calculating taxable income. Due to the different bargaining power of the internal management and ordinary employees, the sharing degree of the company's rent will also be different, which may lead to changes in the internal income gap of enterprises [9,10]. Then, what impact will public welfare donations have on the income of management and ordinary employees? Will it widen or narrow the income gap within enterprises? This study will empirically test and analyze the above problems based on the micro data of A-share listed companies from 2011 to 2020.

The main contributions of this study are as follows: first, it enriches the research on the impact of public welfare donations on the income gap. Most of the existing literature focuses on the positive role of public welfare donations in the three distributions, and few studies focus on the impact of public welfare donations on the income gap within enterprises. Second, it expands the research on the effect of preferential policies of enterprise income tax. The existing literature mainly focuses on the positive effects brought by the preferential policies of enterprise income tax. This study studies the effect of the income tax preferential policies and pre-tax tax deduction of public welfare donations on the internal income gap of enterprises and deepens the research on the effect of tax reduction and fee reduction. Third, it provides some inspiration for improving the compensation system of enterprises and improving the level of corporate governance. Executives can allocate more excess rent through strong bargaining power. If their remuneration is not constrained, it will cause the income gap within the enterprise to widen and affect the operating efficiency of the company.

## Theoretical analysis and research hypothesis

Rent sharing refers to the process in which enterprises pay employees excessive remuneration due to the improvement of business performance, and the result is an increase in employees' average remuneration [11]. In the modern corporate system, company owners consider the growth of business performance as the main indicator for assessing management. In order to maximize their own interests, the management of the enterprise may request to share the enterprise rent [12]. Meanwhile, with the growth of business performance, ordinary employees will also demand to share rent and increase salaries. Otherwise, it will seriously reduce employees' labor enthusiasm and be detrimental to business operations [13]. So, when public welfare donations promote the improvement of business performance, employees (including management and ordinary employees) will share the company's rent.

Public welfare donations increase the rent shared by enterprises and employees in two ways: first, the Pre-tax deduction policy for public welfare donations reduces the tax burden of enterprises, improves the after-tax profits of the company, and thus increases the rent shared by enterprises and employees [14]. Second, public welfare donations can bring many benefits to enterprises while fulfilling their social responsibilities. For example, public welfare donations can receive a good "advertising effect", which can not only reduce the advertising expenditure of enterprises but also improve the brand awareness of enterprises [15]. It can improve the competitive environment of enterprises, facilitate enterprises to obtain resources from external stakeholders [16], significantly reduce the negative impact of adverse factors on the reputation of enterprises, improve core competitiveness and improve financial performance [17], Increase the rent shared by enterprises and employees.

The rent-sharing theory believes that the level of rent-sharing of a company determines the income gap among employees [18]. The bargaining theory points out that the level of salary is determined by the enterprise and employees through bargaining, and is affected by the bargaining power of both parties. Similarly, the amount of rent shared by employees with different bargaining power within the company will also be different, which will lead to the widening of the income gap within the enterprise [19]. Employees of the company can be divided into management and ordinary employees according to their quality. China has a large labor force population base, fierce market competition for ordinary employees, and relatively small supply elasticity; However, as senior managers have higher requirements for academic qualifications, skills, and experience, the number of them is relatively small, and the supply elasticity is relatively large, senior managers can negotiate for high salaries. In addition, managers have the motivation and ability to determine the distribution rules of remuneration and interfere with the remuneration of ordinary employees. When the total amount of remuneration that an enterprise can pay is certain, there is a trade-off relationship between the remuneration of ordinary employees and management. Therefore, this study believes that the bargaining power of management is stronger than that of ordinary employees, so it often occupies an advantage in rent sharing, which ultimately leads to the widening income gap between management and ordinary employees.

To sum up, this study assumes that compared with the per capita remuneration of ordinary employees, the positive impact of the company's public welfare donations on the per capita remuneration of management will be more significant. At the same time, with the increase in corporate public welfare donations, the internal income gap of the company is increasing.

## Research design

**Sample data source.** This study selects A-share listed companies from 2011 to 2020 as the research sample. The sample was screened as follows: (1) Financial and insurance listed

companies were excluded; (2) Excluding ST and *ST listed companies; (3) Exclude listed companies whose per capita remuneration of management is lower than that of ordinary employees, because the amount of management salary disclosed by such companies may only include allowances; (4) The reason for excluding listed companies with negative profits is that such companies cannot examine the tax reduction effect of public welfare donations; (5) Remove samples with incomplete key variables. The data in this study are all from CSMAR. In addition, we also performed tailing on all continuous variables at the upper and lower 1% levels. Finally, 16677 effective samples were obtained.

**Research variables.** *Explained variable. Per capita remuneration of management (GWAGE).* The management studied in this study includes members of the board of directors, members of the board of supervisors, and senior managers [20]. The salary of management personnel mainly includes fixed salary and performance bonus. The scope and proportion of equity payments for Chinese listed companies are relatively small. This study did not consider equity incentives when calculating the average employee salary, but controlled for the management's shareholding ratio in the model. The average remuneration of the management is calculated according to the total annual remuneration and the number of directors, supervisors, and senior executives disclosed in the annual report of the listed company. Among them, the management scale refers to the "total number of directors, supervisors and senior executives" minus the "number of independent directors" and the "number of directors, supervisors and senior executives who are not paid".

*Per capita salary of ordinary employees (YWAGE).* The ordinary employees studied in this study are employees of enterprises other than directors, supervisors, and senior managers [21]. The salary of ordinary employees mainly includes fixed salary and performance bonus. The total salary of ordinary employees is calculated by subtracting the total annual salary of directors, supervisors, and senior managers from the "cash paid to and for employees" in the cash flow statement, and then divided by the number of ordinary employees to get the average salary of ordinary employees.

*Intra-enterprise income gap(GAP)*. The salary data of all employees in the enterprise is difficult to obtain, but the salary gap between the management and ordinary employees of listed companies can be used to replace the income gap between the rich and ordinary. Therefore, this study refers to the methods of Card et al. [10]and uses the salary gap between the management and ordinary employees to represent the income gap within the enterprise. The specific calculation method is as follows: the difference between the per capita salary of the management of the enterprise minus the per capita salary of the ordinary employees of the enterprise is taken as the logarithm.

*Main explanatory variables. Public welfare donation.* According to China's Public Welfare Donation Law, public welfare donations only refer to donations made by enterprises through social organizations, organizations, government agencies, etc. as stipulated by law. This concept is mainly reflected in tax laws. The public welfare donation data in this study comes from the details of "donation expenses", "charitable donation expenses", and "public welfare donation expenses" under the "non operating expenses" item in the notes of listed companies' reports in the CSMAR, which are manually organized and processed. Referring to the method of Zhang et al. [22], the donation amount is logarithmically measured to measure the Donation variable.

*Control variable.* Referring to previous relevant studies[11,20], this study comprehensively considers the possible impact of various factors on the income gap of employees within the enterprise and selects a series of relevant variables that affect the remuneration of senior executives and ordinary employees, including the size of the enterprise (*SIZE*), the integration of two positions (*DUAL*), the net cash flow from operations (*OCF*), the asset-liability ratio (*LEV*), the nature of the enterprise (*SOE*), the time of listing (*AGE*), the Growth rate of operating

**Table 1. Variable definition table.**

| Main variable | Variable name | Variable symbol | Variable measurement method |
|---|---|---|---|
| Explained variables | Internal income gap | GAP | The logarithm of the absolute value of the difference between the per capita remuneration of the management and that of ordinary employees |
| | Management income | GWAGE | Per capita remuneration of management, as defined in the text |
| | Income of ordinary employees | YWAGE | Per capita salary of ordinary employees, as defined in the text |
| Explain variables | Public welfare donation expenditure | Donation | See the text for the definition of enterprise public welfare donation expenditure |
| Control variables | Enterprise scale | SIZE | The logarithm of total assets at the end of the period |
| | Financial leverage | LEV | Total liabilities / total assets |
| | Enterprise nature | SOE | The value of state-owned enterprises is 1, and the value of non-state-owned enterprises is 0 |
| | Years of listing | AGE | Time from research year to market year |
| | Development speed | GROWTH | The growth rate of operating revenue |
| | The shareholding proportion of the largest shareholder | TOP | The shareholding proportion of the largest shareholder (%) |
| | Proportion of independent directors | INBOARD | The proportion of independent directors on the board of directors (%) |
| | Duality | DUAL | Whether the chairman of the board concurrently serves as the general manager = 1, otherwise = 0 |
| | The shareholding ratio of management | MSHARE | The shareholding ratio of management (%) |
| | Cash flow | OCF | Net cash flow from operating activities at the end of the period / total assets at the end of the period |
| Other variables | Tax policy | POLICY | If the year is less than 2017, the value is 0; otherwise, it is 1. |
| | Actual tax negative rate of enterprise | TAXR | Total income tax expense/profit |
| | Enterprise performance | ROE | Net profit / net assets |

revenue (*GROWTH*), the shareholding proportion of the largest shareholder (*TOP*) and shareholding proportion of management (*MSHARE*). As shown in Table 1.

*Research model design*. To test the impact of public welfare donations (Donation) on the per capita remuneration of management (*GWAGE*), model (1) is constructed as follows:

$$GWAGE_{i,t} = \alpha_0 + \alpha_1 Donation_{i,t} + \alpha_2 SIZE_{i,t} + \alpha_3 LEV_{i,t} + \alpha_4 SOE_{i,t} + \alpha_5 AGE_{i,t} + \alpha_6 GROWTH_{i,t}$$
$$+ \alpha_7 TOP_{i,t} + \alpha_8 INBOARD_{i,t} + \alpha_9 DUAL_{i,t} + \alpha_{10} MSHARE_{i,t} + \alpha_{11} OCF_{i,t} + YEAR \quad (1)$$
$$+ IND + \varepsilon_{i,t}$$

To test the impact of public welfare donations (Donation) on the per capita salary of ordinary employees(YWAGE), model (2) is constructed as follows:

$$YWAGE_{i,t} = \alpha_0 + \alpha_1 Donation_{i,t} + \alpha_2 SIZE_{i,t} + \alpha_3 LEV_{i,t} + \alpha_4 SOE_{i,t} + \alpha_5 AGE_{i,t} + \alpha_6 GROWTH_{i,t}$$
$$+ \alpha_7 TOP_{i,t} + \alpha_8 INBOARD_{i,t} + \alpha_9 DUAL_{i,t} + \alpha_{10} MSHARE_{i,t} + \alpha_{11} OCF_{i,t} + YEAR \quad (2)$$
$$+ IND + \varepsilon_{i,t}$$

To test the impact of public welfare donations (Donation) on the internal income gap of enterprises (GAP), a model (3) is constructed as follows:

$$GAP_{i,t} = \alpha_0 + \alpha_1 Donation_{i,t} + \alpha_2 SIZE_{i,t} + \alpha_3 LEV_{i,t} + \alpha_4 SOE_{i,t} + \alpha_5 AGE_{i,t} + \alpha_6 GROWTH_{i,t}$$
$$+ \alpha_7 TOP_{i,t} + \alpha_8 INBOARD_{i,t} + \alpha_9 DUAL_{i,t} + \alpha_{10} MSHARE_{i,t} + \alpha_{11} OCF_{i,t} + YEAR \quad (3)$$
$$+ IND + \varepsilon_{i,t}$$

## Empirical test and analysis

**Descriptive statistics.** It can be seen from Table 2 that the average value of the gap is 1.552 and the standard deviation is 0.614. The difference between the maximum value of 3.313

**Table 2. Descriptive statistics.**

| Variable | Observations | Mean Value | Standard Deviation | Minimum | Median | Maximum |
|---|---|---|---|---|---|---|
| GAP | 16677 | 1.552 | 0.614 | 0.221 | 1.516 | 3.313 |
| GWAGE | 16677 | 13.030 | 0.653 | 11.540 | 13.020 | 14.870 |
| YWAGE | 16677 | 11.480 | 0.496 | 10.290 | 11.470 | 12.770 |
| Donation | 16677 | 0.214 | 0.608 | 0.000 | 0.036 | 4.622 |
| SIZE | 16677 | 22.240 | 1.261 | 20.020 | 22.060 | 26.140 |
| LEV | 16677 | 0.408 | 0.200 | 0.051 | 0.400 | 0.858 |
| SOE | 16677 | 0.329 | 0.470 | 0.000 | 0.000 | 1.000 |
| AGE | 16677 | 9.191 | 7.232 | 0.000 | 8.000 | 26.000 |
| GROWTH | 16677 | 0.299 | 0.750 | -0.601 | 0.122 | 5.148 |
| TOP | 16677 | 34.710 | 14.990 | 8.810 | 32.630 | 75.100 |
| INBOARD | 16677 | 0.382 | 0.071 | 0.250 | 0.364 | 0.600 |
| DUAL | 16677 | 0.289 | 0.453 | 0.000 | 0.000 | 1.000 |
| MSHARE | 16677 | 0.151 | 0.208 | 0.000 | 0.013 | 0.706 |
| OCF | 16677 | 19.280 | 1.564 | 15.260 | 19.220 | 23.500 |

and the minimum value of 0.221 is nearly 14 times, which indicates that the gap among enterprises is large. The average capita remuneration of management (*GWAGE*) is 13.030, while the average per capita salary of ordinary employees (*YWAGE*) is 11.480, with a difference of 1.55, and the standard deviation of the two is 0.653 and 0.496 respectively. This also shows that the per capita salary of the ordinary employees is not as volatile as that of the management, and the per capita salary of the ordinary employees is generally less than that of the management. In addition, the amount of donation income of different enterprises is not the same. The minimum value of public welfare donation is almost 0, the maximum value is 4.622, the average value is 0.214, and the standard deviation is 0.608, which indicates that the public welfare donation of enterprises fluctuates greatly.

**Correlation coefficient analysis of main variables.** *Correlation analysis.* As shown in Table 3, the public welfare donation of enterprises (Donation) is significantly positively

**Table 3. Correlation analysis.**

| | (1) | (2) | (3) | (4) | (5) | (6) | (7) | (8) | (9) | (10) | (11) | (12) | (13) | (14) |
|---|---|---|---|---|---|---|---|---|---|---|---|---|---|---|
| Donation | 1 | | | | | | | | | | | | | |
| GAP | 0.174*** | 1 | | | | | | | | | | | | |
| GWAGE | 0.277*** | 0.689*** | 1 | | | | | | | | | | | |
| YWAGE | 0.154*** | -0.316*** | 0.464*** | 1 | | | | | | | | | | |
| SIZE | 0.428*** | 0.199*** | 0.446*** | 0.341*** | 1 | | | | | | | | | |
| LEV | 0.120*** | 0.043*** | 0.121*** | 0.103*** | 0.546*** | 1 | | | | | | | | |
| SOE | 0.078*** | -0.095*** | 0.035*** | 0.160*** | 0.374*** | 0.325*** | 1 | | | | | | | |
| AGE | 0.128*** | 0.008 | 0.170*** | 0.215*** | 0.463*** | 0.405*** | 0.471*** | 1 | | | | | | |
| GROWTH | -0.004 | -0.055*** | 0.015* | 0.089*** | 0.033*** | 0.091*** | 0.026*** | 0.055*** | 1 | | | | | |
| TOP | 0.084*** | -0.027*** | -0.002 | 0.030*** | 0.162*** | 0.032*** | 0.183*** | -0.089*** | 0.003 | 1 | | | | |
| INBOARD | -0.006 | 0.008 | 0.003 | -0.006 | -0.062*** | -0.067*** | -0.144*** | -0.098*** | -0.009 | 0.021*** | 1 | | | |
| DUAL | -0.049*** | 0.024*** | -0.012 | -0.046*** | -0.191*** | -0.161*** | -0.294*** | -0.257*** | -0.020*** | -0.025*** | 0.095*** | 1 | | |
| MSHARE | -0.112*** | -0.038*** | -0.124*** | -0.116*** | -0.394*** | -0.350*** | -0.485*** | -0.541*** | -0.018** | -0.071*** | 0.140*** | 0.258*** | 1 | |
| OCF | 0.393*** | 0.231*** | 0.446*** | 0.303*** | 0.795*** | 0.354*** | 0.283*** | 0.352*** | -0.021*** | 0.192*** | -0.044*** | -0.148*** | -0.309*** | 1 |

Note: The table shows the Pearson correlation coefficients for the main variables. * * *** And * indicate significant at 1%, 5% and 10% levels, respectively. See Table 1 for the definitions of all variables.

correlated with the per capita remuneration of management (*GWAGE*) at the level of 1%, indicating that the higher the donation expenditure of enterprises, the more the per capita remuneration of management may be, which preliminarily verifies the hypothesis of this study. Secondly, the public welfare donation of enterprises (Donation) is also significantly positively correlated with the per capita salary of ordinary employees (*YWAGE*) at the level of 1%, which indicates that the higher the donation expenditure of enterprises, the more the per capita salary of ordinary employees may be, but the correlation between the donation expenditure of enterprises and the per capita salary of management is lower, which preliminarily verifies the hypothesis of this study. Finally, there is a significant positive correlation between enterprise public welfare donation expenditure (*Donation*) and enterprise internal income gap(*GAP*)at the level of 1%, which indicates that the higher the amount of enterprise public welfare donation expenditure, the larger the enterprise internal income gap, which preliminarily verifies the hypothesis of this study. At the same time, this study also tested the variance expansion factor and found that the mean value of the variance expansion factor was 3.10, which was far less than 10, which showed that there was no serious multicollinearity in the regression.

*Principal regression test and analysis*. First, the impact of corporate public welfare donation expenditure on the per capita remuneration of management. The regression results of model (1) are listed in Item (1) of Table 4. The regression coefficient between the enterprise's public welfare donation expenditure (*Donation*)and the per capita remuneration of management (*GWAGE*) is 0.065, which is significantly positive at the level of 1%, which indicates that the larger the enterprise's public welfare donation expenditure, the higher the management's per capita salary.

Second, the impact of corporate public welfare donation expenditure on the per capita salary of ordinary employees. Item (2) of Table 4 shows the regression results of model (2). The regression coefficient between the public welfare donation expenditure of the enterprise (*Donation*)and the per capita salary of ordinary employees (*YWAGE*) is 0.008, which is not significant, indicating that the excess rent brought by the public welfare donation expenditure does not significantly improve the per capita salary of ordinary employees.

Third, the impact of corporate public welfare donation expenditure on the internal income gap of enterprises. Item (3) of Table 4 shows the regression results of model (3). The regression coefficient between the enterprise's public welfare donation expenditure (*Donation*) and the enterprise's internal income gap (*GAP*) is 0.072, which is significantly positive at the level of 1%, indicating that the larger the public welfare donation expenditure, the larger the enterprise's internal income gap. To sum up, the hypothesis of this study is verified.

## Robustness test

*Replace interpreted variable*. First of all, this study attempts to re-measure the internal income gap of enterprises by taking the logarithm of the ratio of the per capita remuneration of management to the per capita remuneration of ordinary employees and defines this value as *GAP1*. Secondly, this study attempts to re-measure the internal income gap of enterprises according to the ratio of the per capita remuneration of management and the per capita remuneration of ordinary employees and defines this value as *GAP2*. Finally, this study attempts to re-measure the income gap within enterprises according to the Gini coefficient of the per capita remuneration of enterprise management and the per capita remuneration of ordinary employees and defines this value as *GINI*. The calculation formula for the Gini coefficient between management and ordinary employees is: $G_{gy} = \frac{1}{2n^2\mu} \sum_{i=1}^{N} \sum_{j=1}^{N} |a\omega_{gy,\,i} - a\omega_{gy,\,j}|$ Among them, represents the Gini coefficient of inter-group compensation, n represents the number of employees in each group, represents the number of groups, represents the overall average salary of

**Table 4. Gap between public welfare donations and internal income of enterprises.**

| Variables | (1) | (2) | (3) |
| --- | --- | --- | --- |
| | *GWAGE* | *YWAGE* | *GAP* |
| Donation | 0.065*** | 0.008 | 0.072*** |
| | (7.57) | (1.40) | (6.36) |
| SIZE | 0.170*** | 0.068*** | 0.200*** |
| | (25.06) | (13.54) | (21.83) |
| LEV | -0.409*** | -0.255*** | -0.485*** |
| | (-15.25) | (-12.66) | (-13.23) |
| SOE | -0.053*** | 0.138*** | -0.107*** |
| | (-4.45) | (15.83) | (-6.77) |
| AGE | -0.004*** | 0.002*** | -0.008*** |
| | (-5.36) | (3.80) | (-7.20) |
| GROWTH | -0.011* | 0.021*** | -0.024** |
| | (-1.73) | (4.13) | (-2.55) |
| TOP | -0.002*** | 0.001** | -0.003*** |
| | (-7.39) | (2.45) | (-8.15) |
| INBOARD | 0.079 | 0.053 | 0.100 |
| | (1.37) | (1.26) | (1.30) |
| DUAL | 0.027*** | 0.001 | 0.037*** |
| | (2.91) | (0.19) | (3.00) |
| MSHARE | -0.153*** | -0.019 | -0.194*** |
| | (-6.47) | (-1.11) | (-6.20) |
| OCF | 0.088*** | 0.017*** | 0.108*** |
| | (19.32) | (5.13) | (17.59) |
| Constant | 7.385*** | 8.860*** | 6.140*** |
| | (65.56) | (104.57) | (40.40) |
| YEAR | YES | YES | YES |
| IND | YES | YES | YES |
| Observations | 16,677 | 16,677 | 16,677 |
| R-squared | 0.361 | 0.417 | 0.281 |

Note: This table contains the regression results of public welfare donations and the per capita salary of management, the per capita salary of ordinary employees, and the income gap within the enterprise. * * *** And * indicate significance at 1%, 5%, and 10% levels, respectively, and the t-value of the robust standard error is in brackets. See Table 1 for the definitions of all variables.

employees, and represents the per capita salary of the i-th and j-th groups, respectively. The specific regression results are shown in Table 5. Columns (1), (2), and (3) are the regression results of the explained variables GAP1, GAP2, and GINI respectively. The regression coefficients are 0.865, 0.062, and 0.093, which are significantly and positively related to the internal income gap of enterprises at the level of 1%, indicating that the larger the public welfare donation expenditure, the larger the internal income gap of enterprises. The research conclusion of this study is stable.

*Replace explanatory variable*. Remeasure the explanatory variable enterprise public welfare donations, that is, add 1 to the absolute number of public welfare donations to take the logarithm, and put the new indicator *Donation1* back into the regression for testing. The regression results are shown in column (4) of Table 5. The regression coefficient between the public welfare donation expenditure of enterprises (*Donation1*) and the internal income gap of enterprises (*GAP*) is 0.029, which is significantly positive at the level of 1%, indicating that the larger

**Table 5. Replacement of explained variables and explained variables.**

| Variables | (1) GAP1 | (2) GAP2 | (3) GINI | (4) GAP |
|---|---|---|---|---|
| Donation | 0.865*** | 0.062*** | 0.093*** | |
| | (7.98) | (5.65) | (3.44) | |
| Donation1 | | | | 0.029*** |
| | | | | (11.20) |
| SIZE | 0.780*** | 0.101*** | -0.769*** | 0.190*** |
| | (14.12) | (13.60) | (-24.10) | (20.91) |
| LEV | -1.012*** | -0.163*** | -2.331*** | -0.478*** |
| | (-5.29) | (-5.64) | (-16.54) | (-13.06) |
| SOE | -1.314*** | -0.196*** | -0.551*** | -0.100*** |
| | (-13.65) | (-15.09) | (-10.71) | (-6.36) |
| AGE | -0.033*** | -0.007*** | -0.028*** | -0.008*** |
| | (-5.06) | (-7.46) | (-7.55) | (-7.07) |
| GROWTH | -0.177*** | -0.032*** | 0.146*** | -0.024*** |
| | (-3.94) | (-4.44) | (4.19) | (-2.58) |
| TOP | -0.020*** | -0.003*** | -0.015*** | -0.003*** |
| | (-8.17) | (-8.38) | (-10.65) | (-8.12) |
| INBOARD | 0.065 | 0.019 | -1.122*** | 0.108 |
| | (0.15) | (0.31) | (-4.10) | (1.40) |
| DUAL | 0.157** | 0.024** | 0.129*** | 0.039*** |
| | (2.27) | (2.47) | (2.67) | (3.15) |
| MSHARE | -0.926*** | -0.139*** | 0.569*** | -0.206*** |
| | (-5.40) | (-5.57) | (4.08) | (-6.60) |
| OCF | 0.476*** | 0.071*** | -0.013 | 0.106*** |
| | (14.34) | (14.68) | (-0.61) | (17.17) |
| Constant | -15.621*** | -1.449*** | 22.423*** | 6.043*** |
| | (-15.54) | (-11.41) | (43.91) | (41.62) |
| YEAR | YES | YES | YES | YES |
| IND | YES | YES | YES | YES |
| Observations | 16,677 | 16,677 | 16,677 | 16,676 |
| R-squared | 0.189 | 0.169 | 0.297 | 0.285 |

*Note: Items (1) to (3) in this table are the regression results of public welfare donations and remeasured internal income gap (GAP1, GAP2, GINI); Item (4) is the regression result of the remeasured Donation1 and GAP. * * *** And * indicate significance at 1%, 5%, and 10% levels, respectively, and the t-value of the robust standard error is in brackets. See Table 1 for the definitions of all variables.*

the public welfare donation expenditure, the larger the internal income gap. The conclusion of this study is stable.

*Endogenous*. To further account for the presence of time-invariant features over the observation period, we incorporate firm and year fixed effects to control for these influencing factors. The regression results are presented in Table 6. Even after controlling for the firm fixed effects, the coefficient of GAP is 0.019 and significant at 1% level. This suggests that our main regression findings remain robust even after accounting for the firm fixed effects.

Secondly, since the objects tested in this study are only companies whose public welfare donation expenditure has been disclosed and can be collected, this study also conducted the Heckman test to alleviate the problem of self-selection of samples. In the first step of Heckman's two-step regression, the exogenous variable POLICY is added, and the POLICY index

**Table 6. Controlling for the firm fixed effects.**

| Variables | (1) |
|---|---|
| | *GAP* |
| *Donation* | 0.019** |
| | (2.02) |
| *SIZE* | 0.267*** |
| | (22.94) |
| *LEV* | -0.333*** |
| | (-8.06) |
| *SOE* | 0.001 |
| | (0.04) |
| *AGE* | 0.082*** |
| | (2.66) |
| *GROWTH* | -0.029*** |
| | (-4.52) |
| *TOP* | 0.001 |
| | (0.93) |
| *INBOARD* | 0.204*** |
| | (3.27) |
| *DUAL* | -0.008 |
| | (-0.58) |
| *MSHARE* | -0.092* |
| | (-1.76) |
| *OCF* | 0.025*** |
| | (5.68) |
| *Constant* | 5.761*** |
| | (20.33) |
| *YEAR* | YES |
| *FIRM* | YES |
| *Observations* | 16,677 |
| Chi-square | 361.98*** |

Note: This table is the regression results after controlling for the firm fixed effects * * *** And * indicate significance at 1%, 5%, and 10% levels, respectively, and the t-value of the robust standard error is in brackets. See Table 1 for the definitions of all variables.

with a year less than 2017 is assigned 0, otherwise, it is 1. The reason is that China's enterprise income tax law stipulates that the public welfare donation expenditure incurred by enterprises from 2017 is allowed to be deducted except for the part within 12% of the total annual profit; The excess part is allowed to be carried forward for three years for the deduction. The implementation of this policy will directly affect the donation number of enterprises but has no direct impact on the income gap within enterprises.

The regression results are shown in Table 7. In the first stage, probit regression is used. It can be seen that the policy regression coefficient is significantly positive at the level of 1%, and the regression coefficient is 0.091. When the IMR value estimated in the first stage is brought into the second regression, it is found that the regression coefficient between Donation and GAP is still significantly positive at the level of 1%, and the regression coefficient is 0.053, which indicates that the conclusion of this study is still stable after alleviating some endogenous problems.

**Table 7. Heckman test.**

| Variables | (1) | (2) |
|---|---|---|
| | Firststep | Secondstep |
| Donation | | 0.053*** |
| | | (4.29) |
| IMR | | 1.660*** |
| | | (8.51) |
| POLICY | 0.091*** | |
| | (4.65) | |
| SIZE | 0.149*** | 0.332*** |
| | (10.38) | (19.07) |
| LEV | 0.011 | -0.502*** |
| | (0.19) | (-13.37) |
| SOE | -0.283*** | -0.321*** |
| | (-11.23) | (-10.80) |
| AGE | -0.008*** | -0.014*** |
| | (-4.60) | (-10.48) |
| GROWTH | -0.039*** | -0.060*** |
| | (-4.39) | (-6.45) |
| TOP | -0.004*** | -0.007*** |
| | (-6.39) | (-11.96) |
| INBOARD | -0.107 | 0.011 |
| | (-0.80) | (0.14) |
| DUAL | 0.005 | 0.042*** |
| | (0.24) | (3.32) |
| MSHARE | 0.002*** | -0.000 |
| | (3.47) | (-0.42) |
| OCF | 0.062*** | 0.147*** |
| | (6.44) | (19.00) |
| Constant | -3.401*** | 1.974*** |
| | (-15.35) | (3.97) |
| YEAR | YES | YES |
| IND | YES | YES |
| Observations | 22,059 | 16,307 |
| Pseudo R2/R-squared | 0.042 | 0.283 |

Note: This table is the regression result of the Heckman two-step method. * * *** and * indicate significant at 1%, 5%, and 10% levels, respectively, and the t-value of the robust standard error is in brackets. See Table 1 for the definitions of all variables.

## Heterogeneity test

China's state-owned and non-state-owned enterprises differ greatly in salary system and bargaining power of employees, which may lead to differences in the impact of public welfare donations on the internal income gap of different types of enterprises. First of all, the salary system of state-owned enterprises is highly administrative, and the increase of management salary will be strictly restricted; However, the salary system of non-state-owned enterprises is highly market-oriented, and there will be less resistance to increasing the salary of management. Secondly, compared with ordinary employees of non-state-owned enterprises, many employees of state-owned enterprises are in the establishment, facing very low dismissal risk,

and they are large in scale and have a relatively large voice. Therefore, the excess rent brought by public welfare donations may have a greater impact on the internal income gap of non-state-owned enterprises. According to column (1) of Table 8, the regression coefficient of the cross term (Donation x SOE) between the core variable of public welfare donations and corporate nature is -0.175, with a t-value of -8.97, which is significant at the 1% level. The results show that compared with state-owned enterprises, non-state-owned enterprises' public welfare donations have a greater positive impact on the internal income gap of enterprises.

In addition, in the capital market environment, most of the management owns a certain proportion of the shares of the enterprise, which means that the management has the status of both manager and owner. This identity strengthens its voice in sharing the excess rent brought by public welfare donations, that is, in listed companies with higher management shareholding, public welfare donations have a greater positive impact on the internal income gap of enterprises. According to column (2) of Table 8, the regression coefficient of the cross term (Donation x MSHARE) between public welfare donations and management shareholding ratio is 0.347, with a t-value of 4.82, which is significant at the 1% level. The empirical results show that public welfare donations have a greater positive impact on the internal income gap in companies with high management shareholding than those with low management shareholding.

Finally, external supervision is also an important means to promote fair distribution among enterprises. As an information intermediary in the capital market, analyst reports can enhance the transparency of enterprise information and suppress the self-interest behavior of corporate executives. It is reasonable to believe that companies with higher attention from analyst reports will be more transparent and fair in rent distribution. In addition, media coverage can also reflect the unreasonable salary gap in enterprises, triggering public attention and discussion. In order to improve their social image, enterprises will pay more attention to the salary and benefits of employees and actively take measures to narrow the salary gap; It can encourage the government and regulatory agencies to take action, promote the improvement and implementation of relevant laws and regulations, and strengthen the supervision of corporate compensation policies. According to columns (3) and (4) of Table 8, the regression coefficient of the cross term (Donation x ReportAttention) between public welfare donations and research declaration attention is -0.001, with a t-value of -2.08, which is significant at the 5% level. The regression coefficient of the cross term (Donation x MediaAttention) between public welfare donations and media attention is -0.038, with a t-value of -4.78, which is significant at the 1% level. The empirical results indicate that companies facing looser external supervision have a greater positive impact on the internal income gap of public welfare donations.

The above results show that the difference in bargaining power between ordinary employees and management is an important reason for the uneven distribution of excess rent and internal income gap caused by public welfare donations, and further verifies the robustness of the conclusions of this study.

## Mechanism test

To test the intermediary effect in hypothesis deduction. This study defines tax preference as the external rent source of enterprises, defines enterprise performance (*ROE*) as the internal rent source of enterprises, and discusses the intermediary effect of internal and external rents respectively.

## Intermediary effect of enterprise tax preference

Concerning previous studies, the actual tax negative rate (*TAXR*) of enterprises is taken as the measurement index of tax preference [23]. In this study, the actual tax negative rate (*TAXR*) of

**Table 8. Heterogeneity test.**

| VARIABLES | (1) | (2) | (3) | (4) |
|---|---|---|---|---|
| | GAP | GAP | GAP | GAP |
| Donation×SOE | -0.175*** | | | |
| | (-8.97) | | | |
| Donation×MSHARE | | 0.347*** | | |
| | | (4.82) | | |
| Donation×ReportAttention | | | -0.001** | |
| | | | (-2.08) | |
| ReportAttention | | | 0.005*** | |
| | | | (17.39) | |
| Donation×MediaAttention | | | | -0.038*** |
| | | | | (-4.78) |
| MediaAttention | | | | 0.114*** |
| | | | | (8.81) |
| Donation | 0.164*** | 0.055*** | 0.075*** | 0.101*** |
| | (11.14) | (4.55) | (4.11) | (6.50) |
| SIZE | 0.201*** | 0.199*** | 0.138*** | 0.186*** |
| | (22.00) | (21.84) | (13.10) | (19.66) |
| LEV | -0.492*** | -0.490*** | -0.238*** | -0.499*** |
| | (-13.45) | (-13.39) | (-5.56) | (-13.42) |
| SOE | -0.060*** | -0.106*** | -0.095*** | -0.108*** |
| | (-3.66) | (-6.67) | (-5.34) | (-6.75) |
| AGE | -0.008*** | -0.008*** | -0.005*** | -0.007*** |
| | (-7.90) | (-7.09) | (-4.32) | (-6.49) |
| GROWTH | -0.024*** | -0.024** | -0.022** | -0.022** |
| | (-2.59) | (-2.54) | (-2.08) | (-2.31) |
| TOP | -0.003*** | -0.003*** | -0.003*** | -0.004*** |
| | (-7.99) | (-8.01) | (-7.70) | (-8.73) |
| INBOARD | 0.125 | 0.101 | 0.118 | 0.096 |
| | (1.64) | (1.32) | (1.37) | (1.25) |
| DUAL | 0.037*** | 0.037*** | 0.023* | 0.037*** |
| | (2.98) | (2.99) | (1.71) | (3.01) |
| MSHARE | -0.183*** | -0.233*** | -0.317*** | -0.207*** |
| | (-5.86) | (-7.23) | (-8.97) | (-6.48) |
| OCF | 0.107*** | 0.107*** | 0.088*** | 0.105*** |
| | (17.43) | (17.42) | (12.51) | (16.92) |
| Constant | 6.089*** | 6.141*** | 7.703*** | 6.457*** |
| | (40.16) | (40.51) | (41.44) | (39.73) |
| YEAR | YES | YES | YES | YES |
| IND | YES | YES | YES | YES |
| Observations | 16,677 | 16,677 | 12,752 | 16,336 |
| R-squared | 0.285 | 0.282 | 0.315 | 0.287 |

Note: This table shows the regression results of the heterogeneity test of the nature of ownership (SOE) , management shareholding ratio (MSHARE), Report Attention and Media. * * *** And * indicate significance at 1%, 5%, and 10% levels, respectively, and the t-value of the robust standard error is in brackets. See Table 1 for the definitions of all variables.

enterprises is added to the model (3) to test the intermediary effect of tax preference. The regression results are shown in Table 9. The public welfare donation expenditure (Donation) in column (1) hurts the actual tax negative rate (*TAXR*) of enterprises at a significant level of 1%, which indicates that more donations by enterprises will reduce the actual tax burden borne by enterprises. Relying on the tax preference given by the government, enterprises will get more rent to share internally. Column (2) shows that after adding the actual tax negative rate (*TAXR*) of enterprises to the model (3), the regression coefficient between the donation expenditure of enterprises and the internal income gap of enterprises is still significantly positive at the significant level of 1%, which indicates that the intermediary effect of the tax burden

**Table 9. Impact of public welfare donations on tax negative rate.**

| Variables | (1) | (2) |
|---|---|---|
| | *TAXR* | *GAP* |
| *TAXR* | | -0.005*** |
| | | (-7.36) |
| *Donation* | -0.441*** | 0.067*** |
| | (-3.20) | (5.88) |
| *SIZE* | 0.171 | 0.189*** |
| | (1.17) | (19.81) |
| *LEV* | 10.208*** | -0.336*** |
| | (16.00) | (-8.45) |
| *SOE* | 0.044 | -0.090*** |
| | (0.17) | (-5.46) |
| *AGE* | 0.176*** | -0.007*** |
| | (10.49) | (-5.94) |
| *GROWTH* | -0.412*** | -0.034*** |
| | (-3.31) | (-3.51) |
| *TOP* | 0.007 | -0.004*** |
| | (1.24) | (-8.44) |
| *INBOARD* | -0.939 | 0.057 |
| | (-0.82) | (0.72) |
| *DUAL* | -0.122 | 0.038*** |
| | (-0.72) | (2.98) |
| *MSHARE* | -1.127*** | -0.209*** |
| | (-2.84) | (-6.51) |
| *OCF* | -0.485*** | 0.107*** |
| | (-4.84) | (16.41) |
| *Constant* | 9.830*** | 6.413*** |
| | (3.88) | (39.69) |
| *YEAR* | YES | YES |
| *IND* | YES | YES |
| *Observations* | 15,113 | 15,113 |
| *R-squared* | 0.205 | 0.288 |
| *Sobel statistic* | Z = 2.809***(P<0.01) | |
| *Proportion of intermediary effect* | 3.00% | |

Note: This table verifies the intermediary role of the actual tax negative rate (TAXR) of enterprises in public welfare donations and the internal income gap of enterprises. * * *** And * indicate significance at 1%, 5%, and 10% levels, respectively, and the t-value of the robust standard error is in brackets. See Table 1 for the definitions of all variables.

is established, and the result has passed Sobel test, and the amount of intermediary effect is 3.00%. That is, the increase of public welfare donation expenditure of enterprises reduces the tax negative rate, and the decrease of tax negative rate increases the internal income gap of enterprises, so the increase of public welfare donation expenditure of enterprises expands the internal income gap.

## Intermediary effect of enterprise performance

According to the production supply theory, taxation is a cost expenditure for enterprises. Tax incentives can reduce the operating costs of enterprises, keep the output level unchanged while reducing production inputs, and improve the production efficiency of enterprises. According to the theory of economic growth, as operating costs decrease, enterprises can expand their R&D investment, improve their innovation capabilities, and thereby promote the improvement of enterprise performance. Moreover, tax preferential policies can also guide enterprises to invest resources in areas with higher efficiency, which can also promote the improvement of enterprise performance. There are many research results by Chinese scholars on the impact of tax incentives on corporate performance. Empirical results show that tax incentives have a positive impact on corporate technological innovation performance, financial performance and investment performance. The pre tax deduction of public welfare donations, as an important tax preferential policy, can also play a role in increasing corporate performance.

Enterprise performance (*ROE*) is added to the model (3) to test the intermediary effect of internal rent sources. The regression results are shown in Table 10. The public welfare donation expenditure in column (1) has a positive impact on enterprise performance (*ROE*) at a significant level of 1%, which indicates that more donations by enterprises will improve their income, and relying on good operating performance, enterprises will have more rents to share internally. Column (2) shows that after adding enterprise performance (*ROE*) to model (3), the regression coefficient between donation and gap is still positive at the significant level of 1%, which indicates that the intermediary effect of enterprise performance is established, and the result has passed Sobel test, and the amount of intermediary effect is 49.50%. That is, the increase in the company's public welfare donation expenditure improves the company's performance. Due to the different bargaining power of senior executives and ordinary employees, the increase in performance widens the income gap within the enterprise.

## Conclusions and policy implications

This study uses the microdata of A-share listed companies from 2011 to 2020 to investigate the impact of public welfare donations on the internal income gap of enterprises and its mechanism. The results show that public welfare donations significantly improve the per capita salary of management, but have no significant impact on the per capita salary of ordinary employees, thus widening the income gap within enterprises. Consistent with the research of Kong, D. et al. [21], we controlled for industry and annual fixed effects in benchmark regression, thereby controlling for influencing factors that change over time or industry. In addition, to alleviate the impact of endogeneity issues on our observation results, Heckman test and control for individual fixed effects were used. This result also passed a series of robustness tests. However, despite conducting a series of robustness tests, we were unable to rule out the interference of all endogeneity issues [10]. The mechanism test results show that the income tax preference brought by public welfare donations and the improvement of enterprise operating efficiency have increased the excess rent shared by enterprises and employees. Because the management has stronger bargaining power, it shares more excess rent, which leads to the widening of the income gap within the enterprise. Finally, the results of heterogeneity analysis show that the

**Table 10. Impact of public welfare donation on performance.**

| Variables | (1) | (2) |
|---|---|---|
| | *ROE* | *GAP* |
| ROE | | 1.575*** |
| | | (21.68) |
| Donation | 0.014*** | 0.049*** |
| | (11.76) | (4.46) |
| SIZE | -0.014*** | 0.221*** |
| | (-10.25) | (24.46) |
| LEV | -0.098*** | -0.331*** |
| | (-14.90) | (-9.04) |
| SOE | -0.004** | -0.100*** |
| | (-2.15) | (-6.49) |
| AGE | -0.001*** | -0.007*** |
| | (-4.28) | (-6.49) |
| GROWTH | 0.010*** | -0.039*** |
| | (9.21) | (-4.26) |
| TOP | 0.001*** | -0.004*** |
| | (12.13) | (-10.69) |
| INBOARD | 0.013 | 0.079 |
| | (1.39) | (1.04) |
| DUAL | 0.002 | 0.034*** |
| | (1.03) | (2.83) |
| MSHARE | 0.030*** | -0.240*** |
| | (7.88) | (-7.81) |
| OCF | 0.028*** | 0.064*** |
| | (31.57) | (10.04) |
| Constant | -0.137*** | 6.356*** |
| | (-6.35) | (42.43) |
| YEAR | YES | YES |
| IND | YES | YES |
| Observations | 16,677 | 16,677 |
| R-squared | 0.183 | 0.309 |
| Sobel statistic | Z = 13.197***(P<0.01) | |
| Proportion of intermediary effect | 49.50% | |

Note: This table verifies the intermediary role of enterprise performance (ROE) in public welfare donations and the internal income gap of enterprises. * * *** And * indicate significance at 1%, 5%, and 10% levels, respectively, and the t-value of the robust standard error is in brackets. See Table 1 for the definitions of all variables.

internal income gap of non-state-owned enterprises is more affected by public welfare donations, but limiting executive compensation and enhancing employees' bargaining power can alleviate the widening effect of public welfare donations on the internal income gap of enterprises.

Based on the analysis, the following suggestions are made: first, deepen the research on the relationship between preferential policies of enterprise income tax and income distribution, clarify the impact of different types of preferential policies of enterprise income tax on the income gap within enterprises, and provide evidence support for the government to formulate further tax reduction and fee reduction policies. Second, while using the pre–Tax Deduction

Policy of income tax to encourage enterprises to make public welfare donations, we should pay attention to the introduction of supporting measures, develop a reasonable salary incentive system for senior executives, strictly control the disorderly rise of senior executives' salaries, establish a more equitable internal salary distribution system, and achieve the simultaneous reduction of external and internal disparities. Third, the stronger the bargaining power of ordinary employees, the more helpful it is to reduce the impact of public welfare donations on the internal income gap of enterprises, which is particularly obvious in state-owned enterprises. Therefore, in the improvement of the company's governance structure, we should further improve the voice of ordinary employees, give full play to the role of trade unions, and strengthen the supervision and management of employees over the company.

This study also has certain limitations. Firstly, due to the lack of detailed employee salary data published by listed companies, scholars have used the difference between the average salary of management and the average salary of ordinary employees to measure the income gap of enterprise employees. This study also discusses this issue. Future research can use methods such as questionnaire surveys to obtain more detailed statistics on executive and employee compensation in order to draw more accurate conclusions. Secondly, in terms of model design, considering that the rental sharing theory [18] and bargaining power theory [19], we used cannot derive high-order regression results, we did not use a similar inverted U-shaped model [23]. Future research can develop a U-shaped regression model of corporate public welfare donations on internal income inequality based on other theoretical deductions. Finally, although this study adopts the Heckman bipartite method and individual fixed effects model to alleviate the problem of endogeneity, it is undeniable that endogeneity cannot be completely eliminated. Future research can consider mandatory donation regulations implemented under certain specific institutional backgrounds, and use quasi-natural experiments to test the impact of public welfare donations on internal income disparities within enterprises.

## Author Contributions

**Data curation:** Jiantao Chen, Xiao Wang.

**Methodology:** Jiantao Chen, Xiang Luo.

**Writing – original draft:** Jiantao Chen, Xiang Luo.

**Writing – review & editing:** Jiantao Chen, Xiao Wang.

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
