## [Decision Letter · Decision Letter 0]

17 Apr 2023

PONE-D-23-00002Public welfare donation, Rent sharing, and Income gap within enterprisesPLOS ONE

Dear Dr. Luo,

Thank you for submitting your manuscript to PLOS ONE. After careful consideration, we feel that it has merit but does not fully meet PLOS ONE’s publication criteria as it currently stands. Therefore, we invite you to submit a revised version of the manuscript that addresses the points raised during the review process.

ACADEMIC EDITOR: The manuscript provides insight into an interesting topic but to make it more readable and comprehensive, I would like you to carefully consider the reviewers' comments and revise the manuscript accordingly. There is a need to get help from the language expert to make it more readable.Please submit your revised manuscript by Jun 01 2023 11:59PM. If you will need more time than this to complete your revisions, please reply to this message or contact the journal office at plosone@plos.org. Please include the following items when submitting your revised manuscript:A rebuttal letter that responds to each point raised by the academic editor and reviewer(s). You should upload this letter as a separate file labeled 'Response to Reviewers'.A marked-up copy of your manuscript that highlights changes made to the original version. You should upload this as a separate file labeled 'Revised Manuscript with Track Changes'.An unmarked version of your revised paper without tracked changes. You should upload this as a separate file labeled 'Manuscript'.

We look forward to receiving your revised manuscript.

Kind regards,

Yasir Ahmad

Academic Editor

PLOS ONE

Journal Requirements:

b)        State what role the funderstook in the study. If the funders had no role in your study, please state: “The funders had no role in study design, data collection and analysis, decision to publish, or preparation of the manuscript.”

3. PLOS requires an ORCID iD for the corresponding author in Editorial Manager on papers submitted after December 6th, 2016. Please ensure that you have an ORCID iD and that it is validated in Editorial Manager. To do this, go to ‘Update my Information’ (in the upper left-hand corner of the main menu), and click on the Fetch/Validate link next to the ORCID field. This will take you to the ORCID site and allow you to create a new iD or authenticate a pre-existing iD in Editorial Manager. Please see the following video for instructions on linking an ORCID iD to your Editorial Manager account: https://www.youtube.com/watch?v=_xcclfuvtxQ.

Reviewers' comments:

Reviewer's Responses to Questions

**Comments to the Author**

1. Is the manuscript technically sound, and do the data support the conclusions?

Reviewer #1: Yes

Reviewer #2: Yes

2. Has the statistical analysis been performed appropriately and rigorously? 

Reviewer #1: Yes

Reviewer #2: Yes

3. Have the authors made all data underlying the findings in their manuscript fully available?

Reviewer #1: Yes

Reviewer #2: Yes

4. Is the manuscript presented in an intelligible fashion and written in standard English?

Reviewer #1: Yes

Reviewer #2: No

5. Review Comments to the Author

Reviewer #1: Dear Authors,

Thank You for so interesting and relevant Article entitled "Public welfare donation, Rent sharing, and Income gap within enterprises".

1. The article is written on a relevant topic, well-structured, and logically proven.

2. The topic is highly original and relevant in the field.

3. The authors duly describe the subject area compared with other published material.

4. The authors could consider some minor improvements in the logic of the presentation of the results. I'd suggest describing some paragraphs about the Sustainability concept in a separate section Discussion. I'd recommend to move to this new separate section some considerations on limitations and future research from the last section Conclusion.

5. The conclusions are consistent with the evidence and arguments presented.

6. The references are appropriate.

7. Please disclose the abbreviations firstly appeared in the text.

Reviewer #2: 1. I have problems to understand what is meant due to incorrect English, such as CSMAR database should be “China Stock Market Accounting Research Database”. The authors should definitely employ a language editor before the paper gets published.

2. There is the logical contradiction between the first paragraph of “Theoretical analysis and research hypothesis” with the subsequent arguments, such as “when the public welfare donation promotes the improvement of the business performance of the enterprise, employees will share the rent of the enterprise”, especially the paper fails to distinguish the concepts of managers and employees in the first paragraph of “Theoretical analysis and research hypothesis” clearly.

3. Some explanatory variables in the table are labeled incorrectly(the column (4) in table 5, Table 8).

4. the paper should explain the reason of using the fixed effect model to solve what kind of endogenous problem. And the paper needs to verify the applicability of the fixed effects model through Hausman test.

5. The article requires a theoretical analysis of the reasons for the establishment of tax and performance mechanisms.

6. PLOS authors have the option to publish the peer review history of their article (what does this mean?). If published, this will include your full peer review and any attached files.

Reviewer #1: **Yes: **Sergey Barykin

Reviewer #2: No

---

## [Author Response · Author response to Decision Letter 0]

11 Jul 2023

Dear Editor and Reviewers,

We would like to appreciate your effort for having reviewed our submitted manuscript and giving us so many valuable suggestions and helpful comments for improving the manuscript. 

In the revised manuscript, we have improved this paper thoroughly according to your suggestions and comments. And we marked the revised parts in red font.

We are highly appreciated your great support and kind help in improving the quality of this paper. Our detailed responses to all of your individual comments are described in the following parts.

1. I have problems to understand what is meant due to incorrect English, such as CSMAR database should be “China Stock Market Accounting Research Database”. The authors should definitely employ a language editor before the paper gets published. 

Response 1: Thank you for your valuable suggestions. We have hired a language editor to make compliance modifications and edits to the paper.

2. There is the logical contradiction between the first paragraph of “Theoretical analysis and research hypothesis” with the subsequent arguments, such as “when the public welfare donation promotes the improvement of the business performance of the enterprise, employees will share the rent of the enterprise”, especially the paper fails to distinguish the concepts of managers and employees in the first paragraph of “Theoretical analysis and research hypothesis” clearly.

Response 2：This article provides a detailed explanation of the concepts of management and ordinary employees in the dependent variables (1) and (2). In order to provide readers with an understanding of these two concepts when reading the first paragraph, an explanation has been provided in footnote 1 of the first paragraph.

When public welfare donations promote the improvement of business performance, employees will share the rent of the company. "The term" employees "refers to all employees of the company, including management and ordinary employees.

3. Some explanatory variables in the table are labeled incorrectly(the column (4) in table 5, Table 8).

Response 3： Thank you for discovering and pointing out the issue. We have made modifications to the errors in Tables 5 and 8.

4. the paper should explain the reason of using the fixed effect model to solve what kind of endogenous problem. And the paper needs to verify the applicability of the fixed effects model through Hausman test.

Response 4：Due to the presence of certain factors in the model that may not change over time or individuals, these factors are difficult to observe and may be related to disturbance terms, ultimately leading to endogeneity issues. So in order to control for the impact of these factors, we adopted a fixed effects test to alleviate the endogeneity problem of missing variables in the model. In ad dition, this article also validated the applicability of the fixed effects model through the Hausman test, and the test results are listed in the last row of Table 6.

5. The article requires a theoretical analysis of the reasons for the establishment of tax and performance mechanisms.

Response 5： The reasons for tax and performance mechanisms have been added to the text. According to the Production Supply Theory, taxation is a costly expenditure for enterprises. Tax incentives can reduce the operating costs of enterprises, keep the output level unchanged while reducing production inputs, and improve the production efficiency of enterprises. According to the theory of economic growth, as operating costs decrease, enterprises can expand their R&D investment, improve their innovation capabilities, and thereby promote the improvement of enterprise performance. Moreover, tax preferential policies can also guide enterprises to invest resources in more efficient areas, which can also promote the improvement of enterprise performance (Tang Hongxiang et al., 2020). Chinese scholars have made a lot of research results on the impact of tax incentives on corporate performance. The empirical results show that tax incentives have a positive impact on corporate technological innovation performance (Jia Jia, 2017), financial performance (Cai Chang et al., 2017), and investment performance (Yu Guansheng et al., 2023). The pre-tax deduction of public welfare donations, as an important tax preferential policy, can also play a role in increasing corporate performance.

---

## [Decision Letter · Decision Letter 1]

20 Oct 2023

PONE-D-23-00002R1Public welfare donation, Rent sharing, and Income gap within enterprisesPLOS ONE

Dear Dr. luo,

Thank you for submitting your manuscript to PLOS ONE. After careful consideration, we feel that it has merit but does not fully meet PLOS ONE’s publication criteria as it currently stands. Therefore, we invite you to submit a revised version of the manuscript that addresses the points raised during the review process.

I think your manuscript addresses an important area which undoubtedly will be valued for our learned audience. To improve upon your manuscript further, I believe addressing the issues highlighted by the reviewers will improve it to a large extent.

We look forward to receiving your revised manuscript.

Kind regards,

Yasir Ahmad

Academic Editor

PLOS ONE

Journal Requirements:

Reviewers' comments:

Reviewer's Responses to Questions

**Comments to the Author**

1. If the authors have adequately addressed your comments raised in a previous round of review and you feel that this manuscript is now acceptable for publication, you may indicate that here to bypass the “Comments to the Author” section, enter your conflict of interest statement in the “Confidential to Editor” section, and submit your "Accept" recommendation.

Reviewer #1: All comments have been addressed

Reviewer #2: All comments have been addressed

Reviewer #3: (No Response)

 2. Is the manuscript technically sound, and do the data support the conclusions?

Reviewer #1: Yes

Reviewer #2: Yes

Reviewer #3: Partly

 3. Has the statistical analysis been performed appropriately and rigorously?

Reviewer #1: Yes

Reviewer #2: Yes

Reviewer #3: Yes

 4. Have the authors made all data underlying the findings in their manuscript fully available?

Reviewer #1: Yes

Reviewer #2: Yes

Reviewer #3: Yes

5. Is the manuscript presented in an intelligible fashion and written in standard English?

Reviewer #1: Yes

Reviewer #2: Yes

Reviewer #3: Yes

 6. Review Comments to the Author

Reviewer #1: Dear Authors,

Thank You for improving Your manuscript. I can see that the Authors made all requested changes to make the manuscript better.

Reviewer #2: 1.Corporate charitable donation is essentially a way of benefit sharing, and the income distribution gap between management and employees is also a manifestation of corporate social responsibility. Therefore, to some extent, this article discusses the relationship between the fulfillment of social responsibility within and outside the enterprise, or the allocation of economic resources between external and internal responsibilities of the enterprise, The article should appropriately elaborate on the impact of this issue on the research process and findings from the perspective of different forms of social responsibility.

2.The theoretical analysis section in the article involves a variety of theories: principal agent theory, effective wage theory, bargaining theory, The rent sharing theory, which are complicated and overloaded. authors should accurately and concisely use related theory to explain.

Reviewer #3: I think the paper would benefit from a strong discussion section to add to the current 'conclusions and policy' section that needs to tie everything together, with the extant literature, including why this theoretical approach was chosen, whether there are any conflicting theories that were considered, why the fixed effects approach was chosen over other statistical approaches, why these particular robustness or sensitivity tests were done and importantly explain the limitations of this statistical and theoretical approach. While some of these components are presented in the results and introduction it feels a little scattered and not cohesively tied to current literature. I especially could not find the limitations discussed anywhere.

 7. PLOS authors have the option to publish the peer review history of their article (what does this mean?). If published, this will include your full peer review and any attached files.

Reviewer #1: **Yes: **Sergey Barykin

Reviewer #2: No

Reviewer #3: No

---

## [Author Response · Author response to Decision Letter 1]

3 Jan 2024

Dear Editor and Reviewers,

We would like to appreciate your effort for having reviewed our submitted manuscript and giving us so many valuable suggestions and helpful comments for improving the manuscript. 

In the revised manuscript, we have improved this paper thoroughly according to your suggestions and comments. And we marked the revised parts in red font.

We are highly appreciated your great support and kind help in improving the quality of this paper. Our detailed responses to all of your individual comments are described in the following parts.

1. Corporate charitable donation is essentially a way of benefit sharing, and the income distribution gap between management and employees is also a manifestation of corporate social responsibility. Therefore, to some extent, this article discusses the relationship between the fulfillment of social responsibility within and outside the enterprise, or the allocation of economic resources between external and internal responsibilities of the enterprise, The article should appropriately elaborate on the impact of this issue on the research process and findings from the perspective of different forms of social responsibility.

Response 1: Thank you for your valuable suggestion. Corporate social responsibility includes four aspects: economic responsibility, legal responsibility, ethical responsibility, and charitable responsibility. In the introduction of the article, we elaborated on the impact of charitable donations on different forms of social responsibility. In addition, we also emphasize that the corporate social responsibility discussed in this paper specifically refers to charitable responsibility.

2. The theoretical analysis section in the article involves a variety of theories: principal agent theory, effective wage theory, bargaining theory, The rent sharing theory, which are complicated and overloaded. authors should accurately and concisely use related theory to explain.

Response 2：Thank you for your valuable suggestion. This article mainly uses rent sharing theory and bargaining theory, with the specific idea that public welfare donations increase the rent of enterprises. According to the rent sharing theory, both the management and ordinary employees of enterprises require rent sharing. Then according to the bargaining theory, the bargaining power of ordinary employees is weak, while the bargaining power of management is strong, the amount of rent shared by the two will be different, which leads to an widening income gap within the enterprise. We have adjusted the first paragraph in the theoretical analysis and research hypothesis section of the article.

3. I think the paper would benefit from a strong discussion section to add to the current 'conclusions and policy' section that needs to tie everything together, with the extant literature, including why this theoretical approach was chosen, whether there are any conflicting theories that were considered, why the fixed effects approach was chosen over other statistical approaches, why these particular robustness or sensitivity tests were done and importantly explain the limitations of this statistical and theoretical approach. While some of these components are presented in the results and introduction it feels a little scattered and not cohesively tied to current literature. I especially could not find the limitations discussed anywhere.

Response 3: Thank you for your valuable suggestion.We have made changes to the suggestions you made in our conclusions and policies.

Consistent with the research of Kong, D. et al. (2017) [35], we controlled for industry and annual fixed effects in benchmark regression, thereby controlling for influencing factors that change over time or industry. In addition, to alleviate the impact of endogeneity issues on our observation results, Heckman test[43]and control for individual fixed effects[44]were used. This result also passed a series of robustness tests. However, despite conducting a series of robustness tests, we were unable to rule out the interference of all endogeneity issues (Card et al., 2016) 

In terms of research limitations, we have added content on model design. Considering that the rental sharing theory (Zhou W., et al., 2014) and bargaining power theory (Auerbach,A., 2018) we used cannot derive high-order regression results, we did not use a similar inverted U-shaped model (Liu,C. and Gao, J., 2022). Future research can develop a U-shaped regression model of corporate public welfare donations on internal income inequality based on other theoretical deductions.

It should be explained here that we have already controlled for industry annual fixed effects in the benchmark regression, in order to maintain consistency with the core reference articles. To alleviate the interference of endogeneity issues on our observed conclusions, we further utilized individual fixed effects to alleviate endogeneity issues. In this section, we have also added corresponding literature to support and compare for discussion.

---

## [Decision Letter · Decision Letter 2]

20 May 2024

PONE-D-23-00002R2Public welfare donation, rent sharing, and income gap within enterprisesPLOS ONE

Dear Dr. luo,

Thank you for submitting your manuscript to PLOS ONE. After careful consideration, we feel that it has merit but does not fully meet PLOS ONE’s publication criteria as it currently stands. Therefore, we invite you to submit a revised version of the manuscript that addresses the points raised during the review process.

One of the reviewers has provided valuable input for improving the manuscript and I think if you could be able to address these comments in your revision, the paper will be in much shape for the larger audience. I can understand this might require a bit of time but I am sure it will be worth it.

We look forward to receiving your revised manuscript.

Kind regards,

Yasir Ahmad

Academic Editor

PLOS ONE

Journal Requirements:

Reviewers' comments:

Reviewer's Responses to Questions

**Comments to the Author**

1. If the authors have adequately addressed your comments raised in a previous round of review and you feel that this manuscript is now acceptable for publication, you may indicate that here to bypass the “Comments to the Author” section, enter your conflict of interest statement in the “Confidential to Editor” section, and submit your "Accept" recommendation.

Reviewer #2: All comments have been addressed

2. Is the manuscript technically sound, and do the data support the conclusions?

Reviewer #2: Yes

3. Has the statistical analysis been performed appropriately and rigorously? 

Reviewer #2: Yes

4. Have the authors made all data underlying the findings in their manuscript fully available?

Reviewer #2: Yes

5. Is the manuscript presented in an intelligible fashion and written in standard English?

Reviewer #2: Yes

6. Review Comments to the Author

Reviewer #2: The research question of this paper is to examine the impact and mechanism of public welfare donations on the internal income gap of enterprises. The public welfare donations significantly increase the internal income gap of enterprises through the income tax benefits and the increase in corporate operating income. The research design is rigorous. The findings have certain contribution to understanding the corporate public donations and its economic consequence.

The following concerns seem to be further considered.

1.The part of “Heterogeneity test” in this paper seems too simple，the authors could add tests on industry attributes and external supervisory factors, such as media or analysts。

2.The translation marks of the paper are too obvious, especially in the part of “policy implications”. Simultaneously, there are still some wrong description. For example, “the actual tax negative rate (TAXR)”, “China Tai'an (CSMARS) database”, “Guotai An database”.

3.The authors should consider reducing the number of Chinese literature and indicating which ones are Chinese form.

4.The authors should merge and refine the last two paragraphs of the paper, or deleting them directly.

7. PLOS authors have the option to publish the peer review history of their article (what does this mean?). If published, this will include your full peer review and any attached files.

Reviewer #2: No

---

## [Author Response · Author response to Decision Letter 2]

9 Jun 2024

Dear Editor and Reviewers,

We would like to appreciate your effort for having reviewed our submitted manuscript and giving us so many valuable suggestions and helpful comments for improving the manuscript. 

In the revised manuscript, we have improved this paper thoroughly according to your suggestions and comments. And we marked the revised parts in red font.

We are highly appreciated your great support and kind help in improving the quality of this paper. Our detailed responses to all of your individual comments are described in the following parts.

1.The part of “Heterogeneity test” in this paper seems too simple，the authors could add tests on industry attributes and external supervisory factors, such as media or analysts。

Response 1: Thank you for your valuable suggestion. In the heterogeneity test, two factors reflecting external supervision, namely Report Attention and Media Attention, are incorporated. The empirical findings indicate that a more lenient external supervision is associated with a greater positive impact of public welfare donations on the internal income gap of the enterprises.

2.The translation marks of the paper are too obvious, especially in the part of “policy implications”. Simultaneously, there are still some wrong description. For example, “the actual tax negative rate (TAXR)”, “China Tai'an (CSMARS) database”, “Guotai An database”.

Response 2: Thank you for your valuable suggestion. The issues present in the translation of the entire study have been thoroughly revised.

3.The authors should consider reducing the number of Chinese literature and indicating which ones are Chinese form.

Response 3:Thank you for your valuable advice. As the focus of this article is on income gap within Chinese enterprises, We have included all relevant Chinese literature that we have reviewed. In order to enhance the quality of this paper, 21 citations from Chinese literature have been removed, and the remaining 6 citations are numbered 11, 18, 20, 21, 22 and 24.

4.The authors should merge and refine the last two paragraphs of the paper, or deleting them directly.

 Response 4: Thank you for your valuable suggestion. The final two paragraphs of the paper have been revised to address any ambiguity.

---

## [Decision Letter · Decision Letter 3]

11 Jul 2024

PONE-D-23-00002R3Public welfare donation, rent sharing, and income gap within enterprisesPLOS ONE

Dear Dr. luo,

Thank you for submitting your manuscript to PLOS ONE. After careful consideration, we feel that it has merit but does not fully meet PLOS ONE’s publication criteria as it currently stands. Therefore, we invite you to submit a revised version of the manuscript that addresses the points raised during the review process.

Kindly review the comments of reviewer 2 and appropriate modifications need to be made in the manuscript. 

We look forward to receiving your revised manuscript.

Kind regards,

Yasir Ahmad

Academic Editor

PLOS ONE

Journal Requirements:

Reviewers' comments:

Reviewer's Responses to Questions

**Comments to the Author**

1. If the authors have adequately addressed your comments raised in a previous round of review and you feel that this manuscript is now acceptable for publication, you may indicate that here to bypass the “Comments to the Author” section, enter your conflict of interest statement in the “Confidential to Editor” section, and submit your "Accept" recommendation.

Reviewer #2: All comments have been addressed

2. Is the manuscript technically sound, and do the data support the conclusions?

Reviewer #2: Yes

3. Has the statistical analysis been performed appropriately and rigorously? 

Reviewer #2: Yes

4. Have the authors made all data underlying the findings in their manuscript fully available?

Reviewer #2: Yes

5. Is the manuscript presented in an intelligible fashion and written in standard English?

Reviewer #2: Yes

6. Review Comments to the Author

Reviewer #2: 1.There are missing variables " ReportAttention" and "MediaAttention" and the results of their coefficients in columns (3) and (4) of Table 8,repectively.

2.The abbreviation "CSMARS" should be "CSMAR".

3.there are Inconsistent characters in the spaces before each paragraph in this paper.

4.At the end of the paper, the "individual fixed effects model" is mentioned, but in the previous robustness test, individual fixed effect is not controlled. Simultaneouslythe use of the fixed effect model in Table 6 is confusing, the reason and process for its use should be explained.

7. PLOS authors have the option to publish the peer review history of their article (what does this mean?). If published, this will include your full peer review and any attached files.

Reviewer #2: No

---

## [Author Response · Author response to Decision Letter 3]

20 Jul 2024

Dear Editor and Reviewers,

We would like to appreciate your effort for having reviewed our submitted manuscript and giving us so many valuable suggestions and helpful comments for improving the manuscript. 

In the revised manuscript, we have improved this paper thoroughly according to your suggestions and comments. And we marked the revised parts in red font.

We are highly appreciated your great support and kind help in improving the quality of this paper. Our detailed responses to all of your individual comments are described in the following parts.

1.There are missing variables " ReportAttention" and "MediaAttention" and the results of their coefficients in columns (3) and (4) of Table 8,repectively.

Response 1: Thank you for your valuable suggestion. We have included the outcomes of the variables "ReportAttention" and "MediaAttention" along with their coefficients in columns (3) and (4) of Table 8, demonstrating significance at the 1% level.

2.The abbreviation "CSMARS" should be "CSMAR".

Response 2: Thank you for your valuable suggestion. We have corrected the errors in the text.

3.there are Inconsistent characters in the spaces before each paragraph in this paper.

Response 3:Thank you for your valuable advice. We have changed the spaces at the beginning of each paragraph.

4.At the end of the paper, the "individual fixed effects model" is mentioned, but in the previous robustness test, individual fixed effect is not controlled. Simultaneously the use of the fixed effect model in Table 6 is confusing, the reason and process for its use should be explained.

 Response 4: Thank you for your valuable suggestion. To further account for the presence of time-invariant features over the observation period, we incorporate firm and year fixed effects to control for these influencing factors. The regression results are presented in Table 6. Even after controlling for the firm fixed effects, the coefficient of GAP is 0.019 and significant at 1% level. This suggests that our main regression findings remain robust even after accounting for the firm fixed effects.

---

## [Decision Letter · Decision Letter 4]

12 Aug 2024

Public welfare donation, rent sharing, and income gap within enterprises

PONE-D-23-00002R4

Dear Dr. Luo,

We’re pleased to inform you that your manuscript has been judged scientifically suitable for publication and will be formally accepted for publication once it meets all outstanding technical requirements.

Kind regards,

Yasir Ahmad

Academic Editor

PLOS ONE

Additional Editor Comments (optional):

Reviewers' comments:

Reviewer's Responses to Questions

**Comments to the Author**

1. If the authors have adequately addressed your comments raised in a previous round of review and you feel that this manuscript is now acceptable for publication, you may indicate that here to bypass the “Comments to the Author” section, enter your conflict of interest statement in the “Confidential to Editor” section, and submit your "Accept" recommendation.

Reviewer #2: All comments have been addressed

2. Is the manuscript technically sound, and do the data support the conclusions?

Reviewer #2: Yes

3. Has the statistical analysis been performed appropriately and rigorously? 

Reviewer #2: Yes

4. Have the authors made all data underlying the findings in their manuscript fully available?

Reviewer #2: Yes

5. Is the manuscript presented in an intelligible fashion and written in standard English?

Reviewer #2: Yes

6. Review Comments to the Author

Reviewer #2: 1. The abstract needs to be further refined and its research value should be embodied.

2. The citation format of literature in the main text should be consistent（e.g. Conclusions and policy implications)

7. PLOS authors have the option to publish the peer review history of their article (what does this mean?). If published, this will include your full peer review and any attached files.

Reviewer #2: No

---

## [Editor Report · Acceptance letter]

26 Aug 2024

PONE-D-23-00002R4 

PLOS ONE

Dear Dr. luo, 

I'm pleased to inform you that your manuscript has been deemed suitable for publication in PLOS ONE. Congratulations! Your manuscript is now being handed over to our production team.

Kind regards, 

on behalf of

Dr. Yasir Ahmad 

Academic Editor

PLOS ONE